# Anodal transcranial direct current stimulation reduces collinear lateral inhibition in normal peripheral vision

Rajkumar Nallour Raveendran[1,2]*, Katelyn Tsang[2], Dilraj Tiwana[2], Amy Chow[2], Benjamin Thompson[2]

**1** Envision Research Institute, Wichita, Kansas, United States of America, **2** School of Optometry & Vision Science, University of Waterloo, Waterloo, Ontario, Canada

* sachinrajopto@gmail.com

**Data Availability Statement:** All relevant data are within the paper and its Supporting Information files.

## Abstract

Collinear flanking stimuli can reduce the detectability of a Gabor target presented in peripheral vision. This phenomenon is called collinear lateral inhibition and it may contribute to crowding in peripheral vision. Perceptual learning can reduce collinear lateral inhibition in peripheral vision, however intensive training is required. Our aim was to assess whether modulation of collinear lateral inhibition can be achieved within a short time-frame using a single 20-minute session of primary visual cortex anodal transcranial direct current stimulation (a-tDCS). Thirteen observers with normal vision performed a 2AFC contrast detection task with collinear flankers positioned at a distance of 2λ from the target (lateral inhibition) or 6λ (control condition). The stimuli were presented 6˚ to the left of a central cross and fixation was monitored with an infra-red eye tracker. Participants each completed two randomly sequenced, single-masked stimulation sessions; real anodal tDCS and sham tDCS. For the 2λ separation condition, a-tDCS induced a significant reduction in detection threshold (reduced lateral inhibition). Sham stimulation had no effect. No effects of a-tDCS were observed for the 6λ separation condition. This result lays the foundation for future work investigating whether a-tDCS may be useful as a visual rehabilitation tool for individuals with central vision loss who are reliant on peripheral vision.

## Introduction

Peripheral vision is susceptible to a phenomenon called crowding, whereby it is difficult to segregate a target object from other objects that are in close proximity.[1–4] Crowding is a particular concern for patients with macular degeneration who lose central vision and are forced to rely on peripheral vision. These patients often develop a preferred retinal locus (PRL), a specific region of the peripheral retina that is used for fixation.[5–7] Crowding impairs spatial vision at the PRL leading to problems with everyday visual activates such as reading.

Crowding in peripheral vision involves cortical mechanisms that can be modulated. For example, perceptual learning can reduce letter crowding in central vision for observers with

**Funding:** RNR - Envision Postdoctoral research fellowship by LC Industries. BT: NSERC grants RPIN-05394 and RGPAS-477166. The funders had no role in study design, data collection and analysis, decision to publish, or preparation of the manuscript.

**Competing interests:** This research was supported by Envision Postdoctoral research fellowship funded by LC Industries to Rajkumar Raveendran and NSERC grants to Ben Thompson. The funders had no role in study design, data collection and analysis, decision to publish, or preparation of the manuscript. Hence, there is no financial/non-financial competing interest from any of the above-mentioned funders. This does not alter our adherence to PLOS ONE on sharing data and materials.

amblyopia[8,9] and in peripheral vision for observers with macular degeneration[10–13]. However, perceptual learning typically requires a large number of training trials [14] which may be a barrier for patients. In addition, the learning does not always transfer to non-trained stimuli.[15–17] Interventions that can directly modulate mechanisms within visual cortex that may contribute to crowding could complement perceptual learning techniques and enable improved vision in patients with central vision loss.

Transcranial direct current stimulation (tDCS) is a non-invasive brain stimulation technique[18–20] that has the potential to modulate neural mechanisms that contribute to crowding. tDCS involves passing a weak 1-2mA electrical current through two head-mounted electrodes (the anode and cathode) and can induce regional changes in cortical excitability and neurotransmitter concentration that outlast the duration of stimulation. For example, anodal tDCS (a-tDCS) of the motor cortex increases cortical excitability[21] and causes a regional reduction in the concentration of the inhibitory neurotransmitter GABA [22–24]. When applied to the primary visual cortex, a-tDCS modulates contrast sensitivity[25,26], visually evoked potential amplitude[25] and the visual cortex BOLD response[26]. Of particular importance for crowding, a-tDCS can immediately improve Vernier acuity and reduce surround suppression within the near-periphery[27–29], possibly by modulating inhibition within the visual cortex[27].

Lateral masking involves the presentation of a central target Gabor patch between two flanker patches.[12] When the patches have collinear orientation, contrast detection thresholds for the target can be increased (collinear inhibition or lateral masking) or reduced (collinear facilitation) depending on target/flanker separation. Collinear inhibition is distinct from crowding, which involves impaired object recognition rather than elevated detection thresholds. However, collinear inhibition represents a well-established psychophysical technique for assessing low-level inhibitory mechanisms that may contribute to crowding. Collinear inhibition and facilitation arise within the primary visual cortex[30,31]and are present in central [8] and peripheral vision[12,32]. Maniglia et al. [12] observed that collinear lateral inhibition could be reduced by ≈40% in normal peripheral vision after perceptual learning (20 sessions/ week over 8 weeks), indicating that collinear inhibition mechanisms within the periphery are plastic.

In this study, we took a first step towards evaluating whether visual cortex a-tDCS has the potential to reduce crowding in peripheral vision by assessing the acute effects of a single stimulation session on lateral inhibition in participants with normal vision. We observed that 20 min of visual cortex a-tDCS reduced lateral inhibition in the visual periphery of healthy observers by ≈30%. This finding paves the way for future studies designed to evaluate the possibility of inducing long-lasting changes in lateral inhibition using a-tDCS and, in the longer term, the potential use of a-tDCS to reduce crowding in individuals with central vision loss.

## Materials and methods

### Apparatus and visual stimuli

13 participants with best corrected visual acuity of ≤20/20 agreed to participate in the study. All participants provided written, informed consent. The study was approved by the University of Waterloo research ethics committee. All the procedures involved in this research adhered to the tenets of the Declaration of Helsinki. Participants were instructed to fixate a 0.2˚central cross and respond to visual stimuli by pressing a key on a keyboard. Visual stimuli were created using PsychoPy[33,34] (available for free download: http://www.psychopy.org). The stimuli were presented on a 27" LCD (ASUS - https://www.asus.com/ca-en/Monitors/

ROG-SWIFT-PG278QR/) placed at the distance of 50cm from a chinrest. The LCD background luminance was 32 cd/m$^2$.

The visual stimuli consisted of a central target, Gabor patch (visible extent size: 1˚, σ: 5, and spatial frequency: 7 cpd) presented 6˚ to the left of the fixation cross. We selected a spatial frequency of 7 cpd because this spatial frequency produced detection thresholds that were measurable with our 8-bit-luminance-contrast stimulus display system. We presented our stimuli at an eccentricity of 6˚ because about 70% of macular degeneration patients have a scotoma size of <5˚ [5] and this experiment was the first step towards investigating the effects of a-tDCS in individuals with macular degeneration. Two flanker Gabor patches (7 cpd, 0.8 contrast) were presented above and below the target. The flanker Gabor patches were positioned at a distance of 2λ (lateral inhibition) or 6λ (control). Throughout the procedure, fixation was monitored in real-time using an infrared video-based eye tracker (EyeLink II, SR Research, Osgoode, Canada, 500 Hz sampling rate).

## Psychophysical task

The 2AFC lateral masking task involved the measurement of contrast detection thresholds for the central target Gabor patch. The initial contrast of the central Gabor patch was set at 0.5 and its contrast was altered using a 2 down and 1 up staircase procedure with a fixed step size of 0.05. The staircase was terminated either after 50 trials or 5 reversals. The contrast detection threshold was determined as the mean of the last 4 reversals. Two staircases were completed and averaged for each threshold measurement. The same procedure was followed for both flanker distances of 2λ and 6λ. These particular flanker distances were selected based on a previous study reporting collinear inhibition for 2λ but not for 6λ[12] and our own pilot data supporting this observation. In fact, a 6λ flanker distance may induce facilitation.[32,35] The 2λ separation was our experimental condition and the 6λ condition was a control condition to test for any general effects of a-tDCS on contrast sensitivity for the target stimulus. A training session was provided for all participants prior to data collection. Any trials in which eye position deviated by more than 1˚ from fixation were immediately repeated.

## Brain-stimulation (tDCS)

tDCS was delivered by a DC Stimulator MC from NeuroConn gmbh (https://www.rogue-resolutions.com/catalogue/neuro-modulation/dc-stimulator-tes/) using a pair of rubber electrodes (5cm x 5cm) placed inside saline soaked sponges. The electrodes were secured in place by the head strap of the eye tracker over electrode positions Oz (anodal electrode) and Cz (cathodal electrode) identified using the standard 10–20 EEG method (Fig 1). Participants each completed two randomly sequenced stimulation sessions conducted at least 48 hours apart; real 2mA anodal tDCS of the primary visual cortex for 20 minutes and sham tDCS where the current was ramped up and then immediately ramped down with the electrodes kept in place for 20 minutes. Participants were masked to the type of stimulation. During each test session, participants completed a block of four threshold measurements for each flanker distance pre-, during-, 5mins post- and 30mins post-stimulation. The sequence of 2λ and 6λ separation measurements was randomized within each test block.

## Data analysis

Prior to analysis, the standard deviation across reversals was calculated for each individual staircase. Any staircase with a standard deviation of 2 or greater was considered to be unreliable and excluded from analysis. An ANOVA with within-subject factors of stimulation type (anodal vs. sham) and time (pre-, during-, 5 minutes post- and 30 minutes post-stimulation)

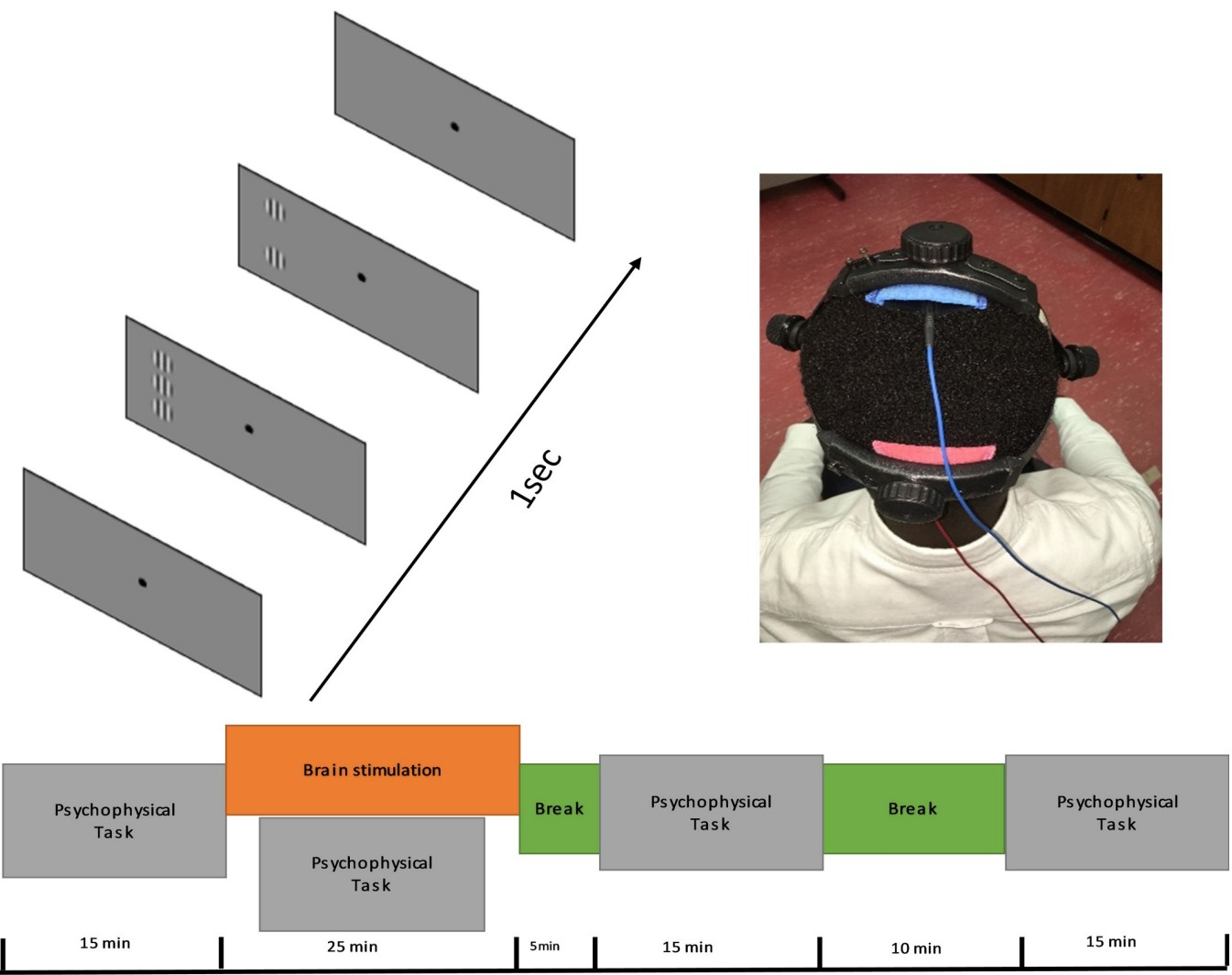

**Fig 1. Experimental design.** A) Collinear configuration of Gabor patches that has the same orientation and phase. B) Sample picture of a participant wearing the eye tracker and the electrodes of tDCS secured using the head strap of the eye tracker. C) Timeline of the experiment. The same timeline was used for anodal and sham stimulations.

was applied to the log contrast thresholds. Post-hoc Tukey HSD was used to compare the log contrast thresholds between different stimulation sessions. Paired t-tests were used to compare session 1 and session 2 baselines to test for task learning. 2λ and 6λ baselines were also compared to ensure that the 2λ separation induced collinear lateral inhibition. A p value of <0.05 was considered statistically significant.

## Results

One staircase had a SD greater than 2 and was removed from further analysis (see S1 and S2 Tables). Baseline contrast detection thresholds for the central Gabor patch differed significantly between the two flanker separation conditions, with higher thresholds for the 2λ separation than the 6λ separation (mean ± SEM; 2λ 0.36±0.03, 6λ 0.14±0.02, $t_{12}$ = 8.2, p < 0.001). This is consistent with previous observations of collinear inhibition within peripheral vision at a target/flanker separation of 2λ[31,32] but not at a 6λ separation. No significant difference

between the baseline measures for session 1 and session 2 were observed for either the 2λ (session 1, 0.39±0.15; session 2, 0.34±0.16) ($t_{12}$ = 1.1, p = 0.30) or 6λ (session 1, 0.13±0.15; session 2, 0.15±0.11) ($t_{12}$ = 0.83, p = 0.42) conditions. This indicates the absence of task learning from one session to the next.

Fig 2 shows raw individual participant data for the a-tDCS and sham stimulation sessions for the 2λ (top row) and 6λ (bottom row) flanker separation conditions (a table of data with mean contrast threshold and standard deviation of staircase reversals for every individual participant is provided in the supporting material: S1 and S2 Tables). Fig 3 shows baseline-normalized group means. For the 2λ separation, a-tDCS significantly improved contrast detection thresholds for the central target (reduced collinear inhibition) whereas sham stimulation had no effect. There was a significant interaction between stimulation type [anodal vs. sham] and time [pre, during, post 5 min, post 30 min] ($F_{3, 36}$ = 3.01, p = 0.042, partial $\eta^2$ = 0.21), with post-hoc Tukey HSD comparisons revealing significantly reduced contrast thresholds relative to baseline for the a-tDCS session during stimulation (p = 0.001) and 30 min post stimulation (p = 0.01). Thresholds at the 5 min post stimulation timepoint did not differ significantly from baseline (p = 0.23). Thresholds within the sham stimulation session did not differ from baseline for any timepoint. For the 6λ separation, there was no significant interaction between stimulation type and time ($F_{3, 36}$ = 0.46, p = 0.71, partial $\eta^2$ = 0.04) indicating no difference between a-tDCS and sham on contrast detection thresholds.

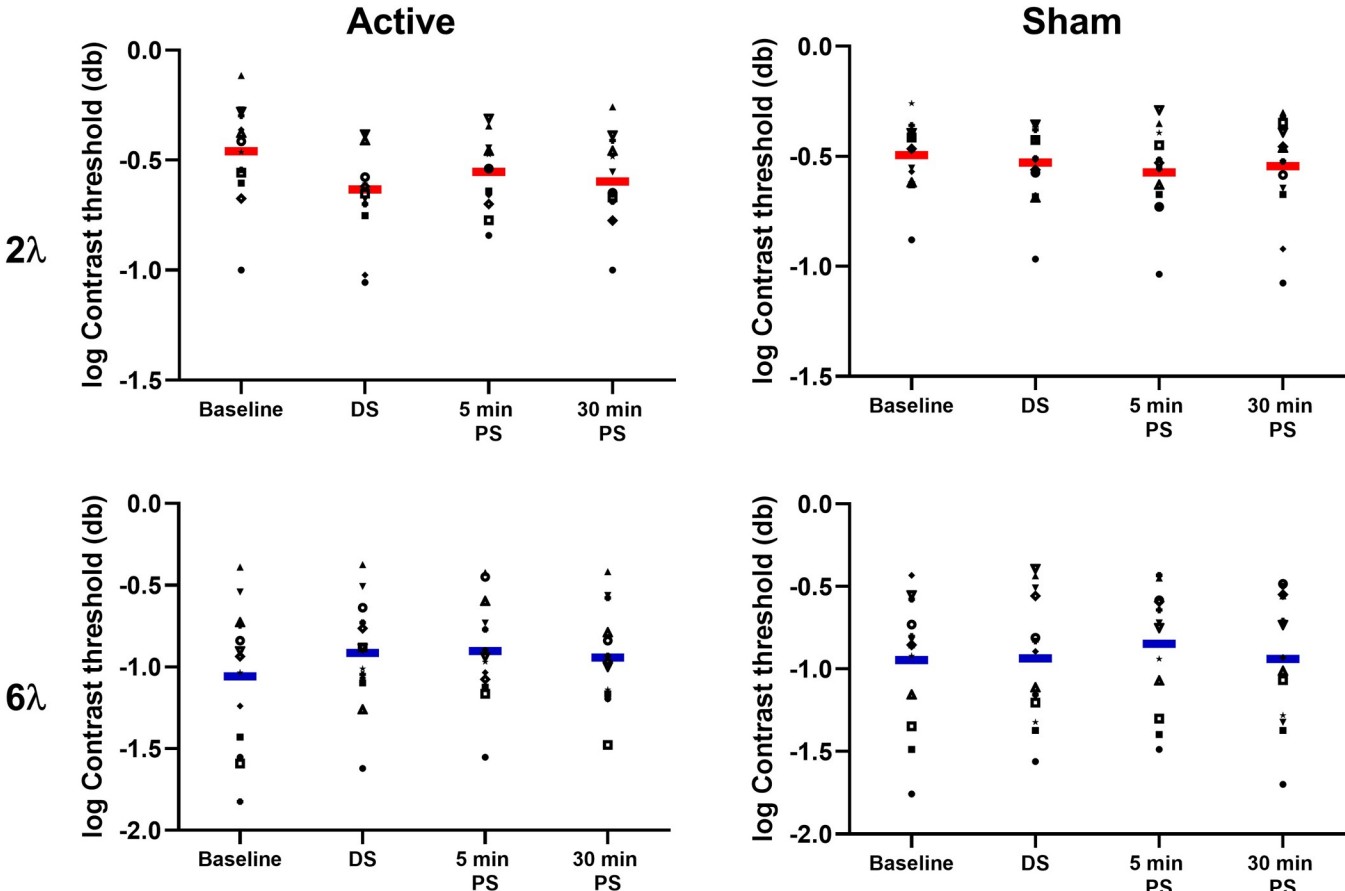

**Fig 2. Contrast threshold.** Log contrast detection thresholds (db) for each participant for the 2λ (top row) and 6λ (bottom row) flanker separations during the active (left column) and sham (right column) stimulation sessions.

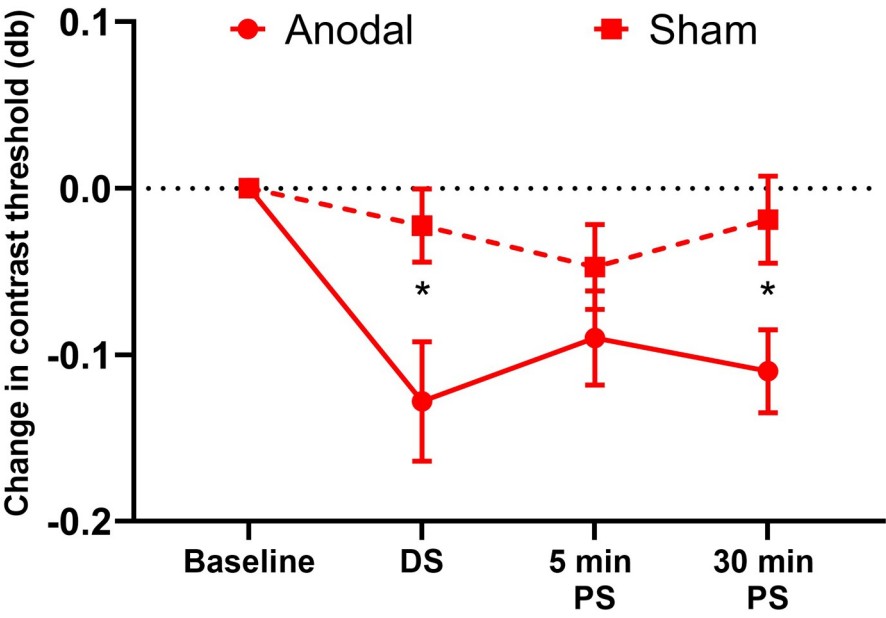

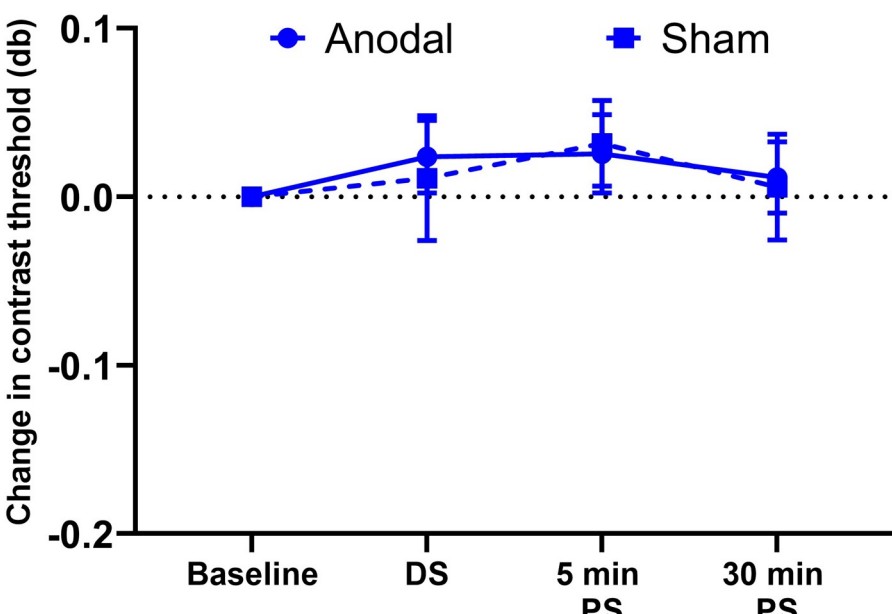

**Fig 3. Reduction of collinear inhibition.** Reduction of collinear inhibition using anodal-tDCS. Mean change in contrast detection threshold from baseline for the 2λ (red) and 6λ (blue) flanker separations for the anodal (solid line) and sham (dashed line) stimulation conditions. Error bars represent ±1 SEM and asterisk symbols represent statistical significance (p<0.05). DS, during stimulation; PS, post stimulation.

## Discussion

The purpose of the study was to test the hypothesis that anodal tDCS would reduce collinear lateral inhibition in peripheral vision of observers with normal vision. The hypothesis was based on previous reports of improved peripheral Vernier acuity, Snellen acuity and contrast sensitivity

[29] along with reduced center-surround suppression[27] following occipital lobe a-tDCS in participants with normal vision. Enhanced contrast sensitivity and modulation of visual cortex activity following a-tDCS have also been observed in patients with amblyopia.[25,26] We observed a significant reduction of collinear inhibition during and 30 min after a single 20 min application of a-tDCS to the occipital lobe. The data collected 5 min post active stimulation exhibited that same trend as data collected during active stimulation and 30 min post active stimulation, but they did not differ significantly from baseline or the sham condition. These results could be likely due to noise inherent in both tDCS effects and psychophysical tasks involving peripheral vision. The effects of a single a-tDCS session are transient, but, overall, our results suggest that a-tDCS is able to modulate low-level lateral interactions in early visual cortex that may contribute to crowding in peripheral vision. The reduction of collinear inhibition that we observed for the measurements made offline (after stimulation) is consistent with previous studies reporting stronger offline than online (during stimulation) primary visual cortex a-tDCS effects [36]. However, in addition to the offline effect, we also observed a significant reduction of collinear inhibition for the online measurements. This is in agreement with a previous report of primary visual cortex a-tDCS effects on surround suppression. In general, offline effects are likely to be more important for the potential use of a-tDCS to improve vision in people with central vision loss as they indicate a lasting influence of a-tDCS on visual cortex function.

A number of explanations have been proposed for the effects of visual cortex a-tDCS. These include changes in response gain[29], stochastic resonance leading to increased signal strength [37], and reduced GABA-mediated inhibition[22,24]. Our results are most clearly aligned with a reduction in cortical inhibition as we observed an effect for the lateral inhibition condition but not the control condition that would have also benefitted from response gain and stochastic resonance changes. Our results also support previous work indicating that lateral inhibition takes place in V1[31,38], the primary target of our stimulation.

Previous studies have observed that collinear lateral inhibition can be reduced using perceptual learning in observers with normal vision[12] and observers with macular degeneration [10]. Maniglia et al.[12] reported an approximately 40% reduction of peripheral collinear lateral inhibition after training in healthy observers (an approximate absolute change in contrast threshold of 0.06). However, in order to achieve this reduction, each participant underwent 160 sessions over the course of 8 weeks ($\approx$ 7600 trials/week). In this study, we observed that a single session of anodal tDCS reduced collinear lateral inhibition by approximately 30% (an absolute change in contrast threshold of 0.13 from baseline to 30 min post active stimulation). This suggests that a-tDCS may enhance the effects of perceptual learning paradigms designed to reduce collinear lateral inhibition. Indeed, a very recent study of healthy adults by Contemori et al. observed that the combination of a different tDCS protocol, transcranial random noise stimulation, and perceptual learning reduced peripheral crowding for trigram stimuli to a greater extent than perceptual learning alone.[39] Furthermore, tDCS increased the transfer of learning to other tasks. Taken together, the present results and the results of Contemori et al.[39] provide a strong foundation for the future application of non-invasive brain stimulation to the rehabilitation of patients with central vision loss, for whom the limitations of peripheral vision represent a major cause of visual disability [40].

One limitation of our study is that there is no consensus on whether lateral masking and crowding are associated, although they share similar features such as increasing strength with eccentricity[41] and a recent study showed that crowding and lateral masking are related and share similar neural mechanisms.[42] In particular, the mechanism of crowding is likely to involve higher visual processing centers.[41] Nonetheless, it is plausible that enhancing the early stages of visual processing by reducing collinear inhibition will improve higher-level visual processing of crowded stimuli.[43,44] In addition, in this study we used a 6λ target

flanker separation distance as a control condition. This separation distance was chosen because it does not induce lateral inhibition, however, flankers at this separation may induce collinear lateral facilitation.[32,35] An alternative control condition could have involved the presentation of orthogonally oriented flankers at a 2λ target-flanker separation.[32,35,45] The fact that we found no effect of a-tDCS for the 6λ separation may suggest that lateral facilitation is not affected by V1 a-tDCS, perhaps because lateral facilitation involves visual areas downstream from V1.[46,47] A study focused directly on lateral facilitation is required to address this interesting possibility. Finally, this study demonstrates only an acute effect of a-tDCS on lateral inhibition. The next stage in the development of this research area will be to study the possibility of long-lasting effects, perhaps by administering multiple a-tDCS sessions[48,49] and/or combining a-tDCS with perceptual learning[50].

## Supporting information

**S1 Table. Mean contrast threshold values (db) and SD of reversals for each participant in 2λ condition.**
(DOCX)

**S2 Table. Mean contrast threshold values (db) and SD of reversals for each participant in 6λ condition.**
(DOCX)

## Acknowledgments

The authors would like to thank Dr. Andy Silva for his assistance with PsychoPy.

## Author Contributions

**Conceptualization:** Rajkumar Nallour Raveendran, Benjamin Thompson.

**Data curation:** Rajkumar Nallour Raveendran, Katelyn Tsang, Dilraj Tiwana, Amy Chow.

**Formal analysis:** Rajkumar Nallour Raveendran, Katelyn Tsang, Dilraj Tiwana, Amy Chow.

**Funding acquisition:** Rajkumar Nallour Raveendran, Benjamin Thompson.

**Investigation:** Rajkumar Nallour Raveendran.

**Methodology:** Rajkumar Nallour Raveendran, Amy Chow, Benjamin Thompson.

**Project administration:** Rajkumar Nallour Raveendran, Amy Chow.

**Resources:** Benjamin Thompson.

**Supervision:** Rajkumar Nallour Raveendran, Benjamin Thompson.

**Validation:** Rajkumar Nallour Raveendran, Katelyn Tsang, Dilraj Tiwana.

**Visualization:** Rajkumar Nallour Raveendran.

**Writing – original draft:** Rajkumar Nallour Raveendran.

**Writing – review & editing:** Rajkumar Nallour Raveendran, Katelyn Tsang, Dilraj Tiwana, Benjamin Thompson.

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
