## [Decision Letter · Decision Letter 0]

25 Mar 2020

PONE-D-20-05503

Anodal transcranial direct current stimulation reduces collinear lateral inhibition in normal peripheral vision

PLOS ONE

Dear Dr. Raveendran,

Thank you for submitting your manuscript to PLOS ONE. After careful consideration, we feel that it has merit but does not fully meet PLOS ONE’s publication criteria as it currently stands. Therefore, we invite you to submit a revised version of the manuscript that addresses the points raised during the review process.

We would appreciate receiving your revised manuscript by May 09 2020 11:59PM. To enhance the reproducibility of your results, we recommend that if applicable you deposit your laboratory protocols in protocols.io, where a protocol can be assigned its own identifier (DOI) such that it can be cited independently in the future. For instructions see: http://journals.plos.org/plosone/s/submission-guidelines#loc-laboratory-protocols

We look forward to receiving your revised manuscript.

Kind regards,

Peter Schwenkreis

Academic Editor

PLOS ONE

Journal Requirements:

"RNR - RNR - LC Industries Postdoctoral research fellowship.

BT: NSERC grants RPIN-05394 and RGPAS-477166.

We note that you received funding from a commercial source: LC Industries

Reviewers' comments:

Reviewer's Responses to Questions

**Comments to the Author**

1. Is the manuscript technically sound, and do the data support the conclusions?

Reviewer #1: Yes

Reviewer #2: Yes

2. Has the statistical analysis been performed appropriately and rigorously? 

Reviewer #1: Yes

Reviewer #2: Yes

3. Have the authors made all data underlying the findings in their manuscript fully available?

Reviewer #1: Yes

Reviewer #2: Yes

4. Is the manuscript presented in an intelligible fashion and written in standard English?

Reviewer #1: Yes

Reviewer #2: Yes

5. Review Comments to the Author

Reviewer #1: Raveendran and colleagues examined whether anodal tDCS applied to the primary visual area decreases collinear lateral inhibition. The authors applied 2mA of sham or anodal tDCS for 20 minutes and found a decrease in collinear lateral inhibition in 2λ distanced Gabor patches, in anodal but not is sham condition - during the stimulation and 30 minutes later. The authors suggest using the findings to induce perceptual training of macular degeneration patients.

The introduction of the manuscript is clear and concise, the method is rigorously explained, the findings as well as real world applications are clear and prominent. The authors elegantly connect between previous findings regarding tDCS effects on early visual processes and the clinical population needs. However, to clarify the implications of the findings and to highlight the necessity for future research some minor revisions are essential.

(1) The manuscript is inconsistent between the broad potential implications and the specificity of the findings. The question presented throughout the introduction is whether MD patients can benefit from a decrease in collinear lateral inhibition as a result of tDCS induced learning protocol - while the experimental manipulation is performed on healthy participants after practice. The findings do not concern perceptual learning alteration, long lasting effects of stimulation or MD patients - these remain yet to be supported hypotheses. Indeed, thirty minutes is a notable timescale for a post-stimulation drag but before suggesting a clinical treatment one should first check the anticipated change at a much greater timescale, the findings are very promising but additional limitations should be mentioned. Additionally, provided that the learning threshold was achieved before the stimulation phase, the tDCS effects are usually considered to be temporarily local (e.g., Antal, Nitsche, & Paulus, 2006). One might think of applying atDCS during the initial practice, so that the early perceptual learning process is strengthened by tDCS. Aiming to improve the visual ability this approach might provide longer lasting effects.

(2) 104-105: tDCS electrode size – most likely the authors meant centimeters and not millimeters.

(3) Punctuation and spacing around reference brackets are inconsistent throughout the text. As the journal does not copyedit the manuscript, it would be beneficial to perfect the typography.

Concluding, the study shows a significant influence of atDCS technique on collinear lateral inhibition and opens a door for future research of tDCS effects on perceptual learning in MD patients.

Reviewer #2: In this paper, the authors tested the hypothesis that transcranial electric stimulation, specifically anodal direct current stimulation (a-tDCS) reduces peripheral collinear inhibition in a single 20-minute session, unlike previous studies that necessitated of multiple behavioral sessions of perceptual learning to achieve this result. Results showed that peripheral collinear inhibition, measured as the contrast threshold for a Gabor target flanked by collinear elements, was indeed reduced during and 30 minutes after a-tDCS. Two control conditions, one in which the flankers were placed at a separation outside the range of collinear inhibition and a second one using sham stimulation, showed no change in contrast thresholds.

The experimental question that the authors tackled is interesting and timely, since collinear inhibition has been linked to visual crowding, one of the main obstacles to object recognition and reading speed in peripheral vision and one of the main targets of rehabilitative therapies for patients suffering from central vision loss.

I find the paper well written and the analysis properly conducted. I would have, however, liked to find some more explanation or discussion of the choice of parameters.

In particular, it would be useful to know the reasons for:

1) The control condition: Authors use a 6λ target-to-flankers separation as a control condition, citing previous studies ‘reporting collinear inhibition for 2λ but not for 6λ’. However, previous studies also showed that 6λ is a separation at which collinear modulation is still present but with an inverse polarity (Maniglia et al. (2015) showed peripheral collinear modulation in peripheral vision up to about 10λ.), rather than just a separation for which collinear inhibition is not present. A ‘cleaner’ approach to this, and a more common control condition in this literature, is the use of orthogonal flankers (Shani and Sagi, 2005; Lev and Polat, 2011; Maniglia et al., 2011) placed at the same target-to-flankers separation. This allows for a control condition that is identical to the experimental condition, with the only difference that no collinear modulation is expected for target-flankers orientation differences of 90° (Polat and Sagi, 1993), thus providing a ‘true’ baseline. It would be important to motivate or comment upon this decision in the paper.

2) The eccentricity of the tested configuration. Not too dissimilar from the previous point, a common eccentricity for the studies of peripheral collinear effect seems to be 4° (Shani and Sagi, 2005; Lev and Polat, 2011; Maniglia et al., 2011; 2015). What was the reason for the authors to choose 6°? Lev and Polat (2011; 2015) suggested that the range of collinear inhibition might increase with eccentricity, thus making the choice of 6λ slightly ‘risky’ as a non-inhibitory separation. However, the different pattern of results seems to solve this point for the authors. Still, only an orthogonal flanker condition would provide a clear reference for the direction of the collinear modulation.

3) The spatial frequency of the stimuli. While foveal studies on collinear facilitation, either single or multiple-session, are usually conducted using mid-high spatial frequencies (e.g., Polat and Sagi, 1993; Polat, 2009), previous studies on peripheral collinear modulation seem to suggest that lower spatial frequencies maximize this effect (Maniglia, Pavan, Trotter, 2015). Why did the authors choose 7cpd?

4) The stimulus size. The size of the stimulus is reported as 1°. In this literature it is common to report the σ of the Gabor patch, with several studies scaling it according to the spatial frequency (often σ=λ, e.g., Polat and Sagi, 1993; 1994; Shani and Sagi, 2005). What was the reason for this specific stimulus size? Previous studies suggest that the interaction between target-to-flankers separation, stimulus size and spatial frequency might affect the range of collinear modulation (Woods, Nugent, Peli, 2002).

Finally, I would comment upon the results at 6λ. Do they suggest that a-tDCS might not affect collinear facilitation (assuming that 6λ at 6° are not too dissimilar from 6λ at 4°)? Indeed, a number of studies suggest that, while collinear inhibition might rely on V1, collinear facilitation might be due at least in part, to extrastriate/higher level feedback mechanisms (Freeman et al. 2001; Angelucci et al., 2002; Angelucci and Bressloff, 2006; Maniglia, Trotter & Aedo-Jury, 2019). The involvement of areas outside the hotspot of the occipital electrode might partially explain this, although it is just a speculation.

Minor points

-Did the participants complete both sessions within the same day? what was the time interval between the two sessions? Did you control for long(er) lasting a-tDCS effects (e.g., difference between PS30_S and BL_A for those who started with real stimulation)?

-What was the luminance resolution?

-Was the luminance of the monitor linearized?

-In the Results section, the thresholds for 6λ should be reported as well (line 135)

-Were the flankers partially superimposed to the target in the 2λ condition? This seems to be a common occurrence for short separation (see Polat and Sagi, 1993, Figure 1). If that was the case, what was the luminance of the overlapping portions?

-A previous paper (Pirulli, Fertonani, Miniussi, 2013) argued that a-tDCS might work best when applied ‘offline’ (i.e., before the behavioral measurement). This seems consistent with your PS30_A results, a little bit less with the DS_A one.

-More of a personal curiosity: It would have been interesting to have crowding measurements before and after stimulation and test whether collinear inhibition and crowding changes correlated.

6. PLOS authors have the option to publish the peer review history of their article (what does this mean?). If published, this will include your full peer review and any attached files.

Reviewer #1: Yes: Taly Bonder

Reviewer #2: Yes: Marcello Maniglia

---

## [Author Response · Author response to Decision Letter 0]

9 Apr 2020

Reviewer #1: Raveendran and colleagues examined whether anodal tDCS applied to the primary visual area decreases collinear lateral inhibition. The authors applied 2mA of sham or anodal tDCS for 20 minutes and found a decrease in collinear lateral inhibition in 2λ distanced Gabor patches, in anodal but not is sham condition - during the stimulation and 30 minutes later. The authors suggest using the findings to induce perceptual training of macular degeneration patients.

The introduction of the manuscript is clear and concise, the method is rigorously explained, the findings as well as real world applications are clear and prominent. The authors elegantly connect between previous findings regarding tDCS effects on early visual processes and the clinical population needs. However, to clarify the implications of the findings and to highlight the necessity for future research some minor revisions are essential.

Thank you very much for your constructive feedback. Please note the following amendments in response to your comments/suggestions. 

(1) The manuscript is inconsistent between the broad potential implications and the specificity of the findings. The question presented throughout the introduction is whether MD patients can benefit from a decrease in collinear lateral inhibition as a result of tDCS induced learning protocol - while the experimental manipulation is performed on healthy participants after practice. The findings do not concern perceptual learning alteration, long lasting effects of stimulation or MD patients - these remain yet to be supported hypotheses. Indeed, thirty minutes is a notable timescale for a post-stimulation drag but before suggesting a clinical treatment one should first check the anticipated change at a much greater timescale, the findings are very promising but additional limitations should be mentioned. Additionally, provided that the learning threshold was achieved before the stimulation phase, the tDCS effects are usually considered to be temporarily local (e.g., Antal, Nitsche, & Paulus, 2006). One might think of applying atDCS during the initial practice, so that the early perceptual learning process is strengthened by tDCS. Aiming to improve the visual ability this approach might provide longer lasting effects.

The point is well taken. We have revised the following sections to address these comments.

• Abstract (Line 33) – “lays the foundation for future work investigating whether”

• Introduction (Page 4; Lines 73-75 & 77-79) - “In this study, we took a first step towards evaluating whether visual cortex a-tDCS has the potential to reduce crowding in peripheral vision by assessing the acute effects of a single stimulation session on lateral inhibition in participants with normal vision… This finding paves the way for future studies designed to evaluate the possibility of inducing long-lasting changes in lateral inhibition using a-tDCS and, in the longer term, the potential use of a-tDCS to reduce crowding in individuals with central vision loss”. 

• Discussion (Page 10; Lines 225 - 228) – “Finally, this study demonstrates only an acute effect of a-tDCS on lateral inhibition. The next stage in the development of this research area will be to study the possibility of long-lasting effects, perhaps by administering multiple a-tDCS sessions and/or combining a-tDCS with perceptual learning”. 

(2) 104-105: tDCS electrode size – most likely the authors meant centimeters and not millimeters.

Thank you for pointing out the error. Now corrected in the revised manuscript (Page: 6; Line: 116)

(3) Punctuation and spacing around reference brackets are inconsistent throughout the text. As the journal does not copyedit the manuscript, it would be beneficial to perfect the typography.

Concluding, the study shows a significant influence of atDCS technique on collinear lateral In this study, we took a first step towards evaluating whether visual cortex a-tDCS has the potential to reduce crowding in peripheral vision by assessing the acute effects of a single stimulation session on lateral inhibition in participants with normal vision.inhibition and opens a door for future research of tDCS effects on perceptual learning in MD patients.

Thank you for pointing out the error. Now corrected in the revised manuscript. 

Reviewer #2: In this paper, the authors tested the hypothesis that transcranial electric stimulation, specifically anodal direct current stimulation (a-tDCS) reduces peripheral collinear inhibition in a single 20-minute session, unlike previous studies that necessitated of multiple behavioral sessions of perceptual learning to achieve this result. Results showed that peripheral collinear inhibition, measured as the contrast threshold for a Gabor target flanked by collinear elements, was indeed reduced during and 30 minutes after a-tDCS. Two control conditions, one in which the flankers were placed at a separation outside the range of collinear inhibition and a second one using sham stimulation, showed no change in contrast thresholds.

The experimental question that the authors tackled is interesting and timely, since collinear inhibition has been linked to visual crowding, one of the main obstacles to object recognition and reading speed in peripheral vision and one of the main targets of rehabilitative therapies for patients suffering from central vision loss.

Thank you very much for your constructive feedback. Please note the following amendments in response to your comments/suggestions.

I find the paper well written and the analysis properly conducted. I would have, however, liked to find some more explanation or discussion of the choice of parameters.

In particular, it would be useful to know the reasons for:

1) The control condition: Authors use a 6λ target-to-flankers separation as a control condition, citing previous studies ‘reporting collinear inhibition for 2λ but not for 6λ’. However, previous studies also showed that 6λ is a separation at which collinear modulation is still present but with an inverse polarity (Maniglia et al. (2015) showed peripheral collinear modulation in peripheral vision up to about 10λ.), rather than just a separation for which collinear inhibition is not present. A ‘cleaner’ approach to this, and a more common control condition in this literature, is the use of orthogonal flankers (Shani and Sagi, 2005; Lev and Polat, 2011; Maniglia et al., 2011) placed at the same target-to-flankers separation. This allows for a control condition that is identical to the experimental condition, with the only difference that no collinear modulation is expected for target-flankers orientation differences of 90° (Polat and Sagi, 1993), thus providing a ‘true’ baseline. It would be important to motivate or comment upon this decision in the paper.

Thank you for this comment. We agree that the suggested control condition would have been appropriate for our study. We chose a 6λ separation for our control condition because we wanted to avoid any potential inhibitory effect of the flankers and we considered facilitatory effects of the flankers to be acceptable. We have now provided a clearer justification for our choice of control condition in the methods section. We have also added the potential use of orthogonal flankers as a control condition to the discussion section as this an excellent suggestion. 

“In addition, in this study we used a 6λ target flanker separation distance as a control condition. This separation distance was chosen because it does not induce lateral inhibition, however, flankers at this separation may induce collinear lateral facilitation” (Page 10 Lines 218-221).

2) The eccentricity of the tested configuration. Not too dissimilar from the previous point, a common eccentricity for the studies of peripheral collinear effect seems to be 4° (Shani and Sagi, 2005; Lev and Polat, 2011; Maniglia et al., 2011; 2015). What was the reason for the authors to choose 6°? Lev and Polat (2011; 2015) suggested that the range of collinear inhibition might increase with eccentricity, thus making the choice of 6λ slightly ‘risky’ as a non-inhibitory separation. However, the different pattern of results seems to solve this point for the authors. Still, only an orthogonal flanker condition would provide a clear reference for the direction of the collinear modulation.

We decided to test at 6° in control participants because about 70% (of 1339 eyes) of patients with macular degeneration (the imminent study group) have scotoma size of <5° (Fletcher, Schuchard, 1997). This is the reason to go beyond the most common 4° eccentricity. We have now justified this design choice in the manuscript 

“We presented our stimuli at an eccentricity of 6° because about 70% of macular degeneration patients have a scotoma size of <5° [5] and this experiment was the first step towards investigating the effects of a-tDCS in individuals with macular degeneration” (Page 5 Lines 94-96).

3) The spatial frequency of the stimuli. While foveal studies on collinear facilitation, either single or multiple-session, are usually conducted using mid-high spatial frequencies (e.g., Polat and Sagi, 1993; Polat, 2009), previous studies on peripheral collinear modulation seem to suggest that lower spatial frequencies maximize this effect (Maniglia, Pavan, Trotter, 2015). Why did the authors choose 7cpd?

Our stimulus display system had a low luminance resolution (8 bit). Therefore, we chose a spatial frequency with detection thresholds that were within the measurable range for our system. Our pilot data indicated that a 7 cpd stimulus produced measurable thresholds and also exhibited lateral inhibition. This is now described in the manuscript as “We selected a spatial frequency of 7 cpd because this spatial frequency produced detection thresholds that were measurable with our 8-bit-luminance-contrast stimulus display system” (Page 5 Lines 92-94).

4) The stimulus size. The size of the stimulus is reported as 1°. In this literature it is common to report the σ of the Gabor patch, with several studies scaling it according to the spatial frequency (often σ=λ, e.g., Polat and Sagi, 1993; 1994; Shani and Sagi, 2005). What was the reason for this specific stimulus size? Previous studies suggest that the interaction between target-to-flankers separation, stimulus size and spatial frequency might affect the range of collinear modulation (Woods, Nugent, Peli, 2002).

1° was the visible extent of our stimulus on the screen. We agree that specifying σ is more appropriate and we have changed this throughout the manuscript (Page 5 Lines 91-94). 

Finally, I would comment upon the results at 6λ. Do they suggest that a-tDCS might not affect collinear facilitation (assuming that 6λ at 6° are not too dissimilar from 6λ at 4°)? Indeed, a number of studies suggest that, while collinear inhibition might rely on V1, collinear facilitation might be due at least in part, to extrastriate/higher level feedback mechanisms (Freeman et al. 2001; Angelucci et al., 2002; Angelucci and Bressloff, 2006; Maniglia, Trotter & Aedo-Jury, 2019). The involvement of areas outside the hotspot of the occipital electrode might partially explain this, although it is just a speculation.

Great comment and a very plausible explanation for the reduced effect of a-tDCS in the 6λ viewing condition. We have added this possibility to the discussion 

“The fact that we found no effect of a-tDCS for the 6λ separation may suggest that lateral facilitation is not affected by V1 a-tDCS, perhaps because lateral facilitation involves visual areas downstream from V1.”(Page 10 Lines 222-224).

Minor points

-Did the participants complete both sessions within the same day? what was the time interval between the two sessions? Did you control for long(er) lasting a-tDCS effects (e.g., difference between PS30_S and BL_A for those who started with real stimulation)?

All participants had at least 48 hours between the sessions (Page 6; Line 120). We randomized the testing sequence to average out any potentially confounding between-session effects (Page 7; Line 146-147). 

-What was the luminance resolution?

8-bit, now specified in the manuscript (Page 5; Line 93)

-Was the luminance of the monitor linearized?

Yes, the screen was Gamma calibrated (γ = 2.2). 

-In the Results section, the thresholds for 6λ should be reported as well (line 135)

Yes, thank you for spotting it. The values have been added now. Page-7; Line 147.

-Were the flankers partially superimposed to the target in the 2λ condition? This seems to be a common occurrence for short separation (see Polat and Sagi, 1993, Figure 1). If that was the case, what was the luminance of the overlapping portions?

No, the target and flankers were not overlapping (but they were close).

-A previous paper (Pirulli, Fertonani, Miniussi, 2013) argued that a-tDCS might work best when applied ‘offline’ (i.e., before the behavioral measurement). This seems consistent with your PS30_A results, a little bit less with the DS_A one.

This is an interesting point and we now mention this possibility in the discussion and include the suggested reference

“The reduction of collinear inhibition that we observed for the measurements made offline (after stimulation) is consistent with previous studies reporting stronger offline than online (during stimulation) primary visual cortex a-tDCS effects. However, in addition to the offline effect, we also observed a significant reduction of collinear inhibition for the online measurements. This is in agreement with a previous report of primary visual cortex a-tDCS effects on surround suppression. In general, offline effects are likely to be more important for the potential use of a-tDCS to improve vision in people with central vision loss as they indicate a lasting influence of a-tDCS on visual cortex function.” (Page 8; Line 184-190). 

-More of a personal curiosity: It would have been interesting to have crowding measurements before and after stimulation and test whether collinear inhibition and crowding changes correlated.

We agree and this a question that our group is currently working on.

---

## [Editor Report · Decision Letter 1]

13 Apr 2020

Anodal transcranial direct current stimulation reduces collinear lateral inhibition in normal peripheral vision

PONE-D-20-05503R1

Dear Dr. Raveendran,

We are pleased to inform you that your manuscript has been judged scientifically suitable for publication and will be formally accepted for publication once it complies with all outstanding technical requirements.

With kind regards,

Peter Schwenkreis

Academic Editor

PLOS ONE
---

## [Editor Report · Acceptance letter]

15 Apr 2020

PONE-D-20-05503R1 

Anodal transcranial direct current stimulation reduces collinear lateral inhibition in normal peripheral vision 

Dear Dr. Raveendran:

I am pleased to inform you that your manuscript has been deemed suitable for publication in PLOS ONE. Congratulations! Your manuscript is now with our production department. 

With kind regards,

on behalf of

Dr. Peter Schwenkreis 

Academic Editor

PLOS ONE